# Accurate Low Complexity Quadrature Angular Diversity Aperture Receiver for Visible Light Positioning

**DOI:** 10.3390/s24186006

**Published:** 2024-09-17

**Authors:** Stefanie Cincotta, Adrian Neild, Kristian Helmerson, Michael Zenere, Jean Armstrong

**Affiliations:** 1Manufacturing Research Unit, Commonwealth Scientific and Industrial Research Organisation (CSIRO), Clayton, VIC 3168, Australia; stef.cincotta@csiro.au; 2Department of Electrical and Computer Systems Engineering, Monash University, Clayton, VIC 3800, Australia; michael.zenere@monash.edu; 3Department of Mechanical and Aerospace Engineering, Monash University, Clayton, VIC 3800, Australia; adrian.neild@monash.edu; 4School of Physics and Astronomy, Monash University, Clayton, VIC 3800, Australia; kristian.helmerson@monash.edu

**Keywords:** indoor positioning system (IPS), visible light positioning (VLP), angle of arrival (AOA), quadrature angular diversity aperture receiver (QADA), programmable system on a chip (PSoC), sensor fusion

## Abstract

Despite the many potential applications of an accurate indoor positioning system (IPS), no universal, readily available system exists. Much of the IPS research to date has been based on the use of radio transmitters as positioning beacons. Visible light positioning (VLP) instead uses LED lights as beacons. Either cameras or photodiodes (PDs) can be used as VLP receivers, and position estimates are usually based on either the angle of arrival (AOA) or the strength of the received signal. Research on the use of AOA with photodiode receivers has so far been limited by the lack of a suitable compact receiver. The quadrature angular diversity aperture receiver (QADA) can fill this gap. In this paper, we describe a new QADA design that uses only three readily available parts: a quadrant photodiode, a 3D-printed aperture, and a programmable system on a chip (PSoC). Extensive experimental results demonstrate that this design provides accurate AOA estimates within a room-sized test chamber. The flexibility and programmability of the PSoC mean that other sensors can be supported by the same PSoC. This has the potential to allow the AOA estimates from the QADA to be combined with information from other sensors to form future powerful sensor-fusion systems requiring only one beacon.

## 1. Introduction

The urgent need for a practical, accurate indoor positioning system is not in question. The Global Navigation Satellite System (GNSS) is an important part of modern life, but because GNSS is often not available indoors or, when available, may not be accurate enough, the use of GNSS is largely restricted to outdoor applications. The clear and unmet need for an accurate indoor positioning system (IPS) has resulted in a large and rapidly increasing international research effort devoted to this problem [1,2,3]. Many different underlying technologies have been explored, and many algorithms developed, yet no generally available system has emerged. Of the many different system structures that have been considered perhaps the simplest conceptually is that used in the GNSS: a number of different beacons each transmit signals from these the receiver calculates its distance from the beacon. The signals also contain data about the position of the transmitting beacon. By combining this information, the receiver calculates its position using trilateration. 

Much of the early research on IPS was devoted to radio frequency (RF) based systems with a structure similar to that of GNSS. Unlike GNSS, which estimates distance based on time difference of arrival (TDOA), most of the research on RF IPS systems has considered the use of received signal strength (RSS) to estimate distance. RSS estimation depends on knowledge of *both* the relationship between attenuation and distance *and* the power of the transmitted signal. The major challenge in RF systems is the effect of multipath: the signals reaching the receiver may consist of both direct and reflected signals. This makes the relationship between attenuation and distance unpredictable. Another approach is to estimate the position of the receiver using angle-of-arrival (AOA) estimation and triangulation. The underlying assumption is that the AOA indicates the direction of the transmitting beacon. Unfortunately, in RF systems, multipath also limits AOA estimation, as the addition of reflected components changes the AOA of the resultant received signal.

More recently, a new IPS technology has emerged [4,5,6,7]. The widespread introduction of LED lighting has paved the way for visible lighting positioning (VLP). LED luminaires, unlike legacy fluorescent and incandescent lighting, can be modulated at high frequencies and hence transmit high-speed data. As a result, they can be used as transmitters for visible light communications (VLC) and/or as VLP beacons. Over the last decade, there has been an exponentially increasing number of papers devoted to VLP [8]. VLP receivers can be divided into two broad classes: camera-based receivers and photodiode (PD)-based receivers. While camera-based systems can provide very accurate AOA estimates, the ability of cameras to receive and decode a modulated signal is very limited [8]. Photodiodes, on the other hand, can act as very high data rate receivers.

Most of the research on PD VLP systems has considered the use of RSS, but while reflections have much less effect in VLP than RF, the problem remains that the power of the optical signal transmitted in the direction of the receiver is unknown. It depends both on the radiation pattern of the luminaire, which is in general not precisely specified, and the intensity of the transmitted light, which reduces with time as the LEDs age. In contrast, AOA has many advantages: the AOA accurately represents the direction of the luminaire, so the only requirement of a beacon is that it knows its position and can transmit this information. There is comparatively little published work on AOA VLP using PD receivers, in part because of the mistaken assumption that AOA receivers are inevitably bulky and complex.

There is a small but growing literature on AOA receivers using PDs. Three broad categories of receivers can be discerned: (1) receivers using three-dimensional structures with PDs pointing in different directions [9,10,11,12,13,14]; (2) receivers using a lens to create an image on a position-sensitive detector (PSD) [15]; and (3) receivers using an aperture or apertures above a number of PDs located on a plane [16,17,18,19,20,21]. All three types have been shown to work, but accurate three-dimensional structures may be difficult to manufacture and will be difficult to integrate into future smartphones. Lens-based systems are possible, but it is not clear what advantages they have over aperture-based systems, which are inherently compact and planar.

The use of aperture receivers for VLC was first proposed in [22]. This used multiple apertures and multiple separate PDs. Its use in VLP was described in [23]. In [16], a new, much simpler receiver design, the quadrant photodiode angular diversity aperture receiver (QADA), was described. This used a square aperture and a square quadrant photodiode (QPD). Its simple structure makes the QADA compact, easy to manufacture, and compatible with incorporation in future smart devices. Since then, there has been a small but growing body of both experimental and theoretical work on QADA VLP systems. A number of the papers use a circular rather than a square aperture [21,24], and some use infrared LEDs as beacons rather than visible light LEDs [24], but the basic principle is the same.

Although lens-based PSD systems and aperture-based AOA systems have some aspects in common, they operate in a fundamentally different way. This can best be explained by considering the simple example of a single small transmitting LED luminaire. The lens in the PSD receiver creates a focused image on the PSD so that, ideally, the luminaire results in a small illuminated spot. The aim of signal processing in this system is then to *estimate the position of this small spot* [25]. In contrast, for a QADA, this luminaire would illuminate a significant proportion of the QPD, not just a small spot. The aim, in this case, is *to estimate the centroid of the illuminated area.* Common misconceptions, such as the QADA being a form of the pinhole camera or that the performance of a QADA is limited by diffraction effects caused by the aperture, may stem from a misunderstanding of this difference. (The operation of a pinhole camera depends on the distance from the aperture to the imaging plane being of the order of one hundred times the diameter of the aperture.) (Diffraction would only be important if the diameter of the aperture was of the same order of magnitude as the wavelength of the light.)

As a result of the limitations of ISP systems based on single technologies, there has been a growing body of research on multi-sensor fusion, in which information from multiple different sensors is combined [1]. In parallel with this has been the development of increasingly sophisticated algorithms to process the resulting data [1]. More recently, research on VLP has followed the same trends [6,26], but so far, there are only a few papers in which information from other sensors is combined with an aperture receiver. In [27], ultrasonic sensing was combined with an infrared QADA sensor, and in [19], we described the hybrid imaging photodiode (HIP) receiver, which combines a QADA with a camera-based sensor.

In this paper, we describe a new compact design for a QADA based on a readily available system on a chip. Extensive experimental results demonstrate that the new design can provide good AOA estimates and reliably identify the direction of a transmitting luminaire. The sources of errors in the estimates are investigated in detail and found to result mainly from limitations in the manufacture of the aperture. The availability of this versatile and practical AOA receiver opens up many important avenues for future research. One important possibility lies in sensor fusion, which integrates the QADA with other sensors. This will enable accurate 3D positioning even when only a single luminaire beacon is within the field of view [FOV]. For the case of a moving receiver, estimates from different positions may also be used to improve the accuracy of position estimates. The study of systems using advanced algorithms to combine the input from multiple sensors and information from different luminaires and then apply sophisticated signal processing algorithms offers a wealth of promising opportunities for future research. These have the potential to lead to the development of new high-performance positioning systems. Although this paper concerns visible light systems, the techniques and insights are equally applicable to systems using infrared beacons.

## 2. Description of QADA Receiver

In this section, we describe the new receiver design and explain its operation and basic properties. The prototype QADA receiver is shown in Figure 1. It was constructed on a standard breadboard and consists of three readily available parts: a programmable system-on-a-chip (PSoC) PSoC5LP prototyping board manufactured and available from Infineon [28], an off-the-shelf Hamamatsu S5980 QPD available from Hamamatsu, Iwata City, Japan [25], and a 3D-printed aperture. The PSoC includes both digital and analog components and thus enables systems using both digital and analog components to be rapidly designed and tested. The S5980 QPD [29] has a common cathode construction with four square segments, each 2.5 × 2.5 mm. The aperture was designed to fit tightly over the QPD and was printed on a Tiertime Upbox [30] using a 50-micron layer resolution with black filament.

Figure 2 shows the design that was loaded into the PSoC and the (external) QPD. Each of the four quadrants of the photodiode and its common cathode are directly connected to the PSoC; no other electronic components are required.

The PSOC prototyping board and the associated, freely downloadable software PSoC Creator 4.4 allow hardware and firmware to be developed and loaded onto the PSoC. Figure 1 was created within PSoC Creator. The PSoC section of Figure 1 shows the PSoC components used in the prototype receiver. Each of the components was drag-and-dropped from the PSoC component catalog, and the graphical interface was used to connect the appropriate terminals. During the design process, each component was configured with the parameters to be used when power was applied to the PSoC. For example, each of the four IDAC8 current sources was configured to sink 1 uA of current. Each transimpedance amplifier (TIA) was configured with a 500 kΩ feedback resistor. The associated default values of feedback capacitor and corner frequency were not modified. The successive approximation analog-to-digital converter (ADC) was initialized to a resolution of 12 bits, an input range of 0 to 2.048 volts, a conversion rate of 62,500 samples per second, and set to software triggering. The USB component has many configurable options, but this design used the PSOC default setting for the USBUART. The transimpedance amplifiers (TIAs), analog multiplexers, and ADC within the PSoC process the four input signals from the QPD. Each quadrant is multiplexed on a sample-by-sample basis to ensure that the samples from different quadrants are taken as synchronously as possible.

The firmware is written as a C program within the PSoC Creator environment, which provides many built-in functions. The program used in the experiments described in this paper was very basic. After the power on initialization, it just looped continuously, waiting for a character to be received on the USBUART (from the host computer) and then output a preset number of sample values as characters via the USBUART to the host computer.

To minimize quantization errors, the PSoC configuration was carefully designed so that the dynamic range of the receiver matched the range of signals occurring within the experiments. This was achieved by jointly optimizing the current and voltage offsets at the inputs to the transimpedance amplifiers, the gain of the amplifier, and the configuration of the ADC. The resulting system is extremely sensitive: for the design in Figure 2, a one-bit change at the output of the ADC represents a 1 nA change in the input current from the QPD quadrant. To optimize performance, all the voltage and current sources were referred to the internal temperature-compensated PSoC voltage reference. In the experiments, the USB provided power to the receiver, but the power consumption of the system was tiny, meaning that a QADA within a cell phone could be powered directly from the phone without significant battery drain. A minor disadvantage of the design, which is caused by the limited number of amplifiers available in this PSoC, is that a *decreasing* received light intensity results in an *increasing* digital output.

In the experiments, the digital values were downloaded via the USB for offline processing. Because of the memory limitations of the PSoC, samples were captured in batches of 20,000 comprising 5000 per quadrant before downloading. There are many possible ways of capturing and processing this information. In the experiments, all of the processing was performed on one computer using a sequence involving CoolTerm version 1.9.0, Microsoft Excel version 2108 and MATLAB version R2022b. CoolTerm software can be downloaded for free from the internet. It allows data received on the USB port to be saved in a timestamped file. Each transfer of data from the PSoC was initiated by pressing any key on the computer keyboard. An Excel spreadsheet was used to store the address of each file of captured data, along with all of the associated parameters related to the characteristics and position of the QADA and the luminaire for that capture. The values were then imported to MATLAB using the import function. Downloading the data and processing them with MATLAB allows a detailed analysis of the sensor performance, but it should be noted that the calculation of the angle-of-arrival (AOA) of the light within the PSoC is also possible. Figure 1 shows two USB connections to the PSOC: one is used for downloading data, and the other for programming. The programming connection is required only during initial programming, not during the experiments.

The operation of the QADA depends on measuring the *ratio* of the signal output from each quadrant, so it is important that the front ends of the circuits are well-matched, as any difference may contribute to errors in angle estimates. To measure the mismatch, captures were first made with the aperture removed and with no light source in the test chamber. In this case, any differences between quadrants can be attributed to a mismatch. The results for a typical capture of 5000 samples per quadrant are shown in Figure 3. Figure 3 also shows the mean and standard deviation for each of the quadrants.

The difference between the mean values of quadrants represents less than 3% of the 0 to 4096 range of the 12-bit ADC. The standard deviation is also around 3%. The averages were very consistent between captures, but slight changes in the standard deviation were observed as the circuit warmed up.

It is important that the quadrants are also well-matched across the range of optical signals occurring in the experiments. To check this, we used a luminaire transmitting a square-wave light signal, as described in the next section. Figure 4 shows the histograms of the resulting received signal and the average and standard deviation for ‘high’ sample values and ‘low’ sample values. Because of the finite rise time of the transmitted optical signal, about 2% of samples were not in either the ‘high’ or the ‘low’ range. These results demonstrate that the quadrants and associated circuitry are well-matched and that the differences in the averages of high and low levels are *extremely* small, within 0.2% of the range of the ADC. This is important as it is these differences that are used to estimate the received signal powers in the calculation of AOA.

## 3. Test Environment

### 3.1. Test Chamber

A specially designed test chamber with internal dimensions of 3 m × 3 m × 2.55 m was built to enable the QADA prototype to be evaluated experimentally. The frame was constructed from T-slot aluminum with moveable crossbeams. This enables luminaires to be mounted at any position within the chamber. To exclude external light, the top and sides of the chamber were clad with corflute. To minimize reflections, a matt black corflute was chosen. The floor is covered with a 3 m × 3 m vinyl mat, which is printed with a grid of lines at 50 mm spacing. (The vinyl mat was a specially commissioned banner from Gorilla Print, a banner manufacturer.) For ease of positioning, grid lines at 500 mm spacing are colored red, and grid lines at 250 mm spacing are colored blue, with the remaining lines colored black. The receiver is placed on a custom-built test platform, which has a 500 mm × 500 mm base and top. The height of the platform can be adjusted in 100 mm increments between 100 mm and 1600 mm by adding vertical stages. The use of the platform ensures that the receiver is always level and allows the receiver to be quickly and accurately positioned and aligned with the floor grid. Figure 5 shows the platform, the QADA prototype, the test chamber, and the vinyl floor mat.

Two coordinate frames are used in this paper: a room-based one $\left[X_{R},Y_{R},Z_{R}\right]$ and a QADA-based one $\left[X_{Q},Y_{Q},Z_{Q}\right]$. The room coordinate frame is aligned with the floor grid and has an origin at the grid intersection at one corner of the test chamber.

### 3.2. Transmitting Luminaire

The luminaire used in the experiments is a 240 mm × 240 mm square panel luminaire. This is used so that we can demonstrate the suitability of the QADA as the first stage of a hybrid receiver, as described in [18,19,20]. The AC drive components were removed from the luminaire so that the LEDs could be directly modulated. A PSoC controls the signal to the luminaire. This allows a variety of signals to be generated, including square waves of programmable frequency and ASCII-coded signals, which can be used to transmit information about the position and characteristics of the luminaire. In the positioning experiments, a square wave was used to modulate the light. Ideally, for experimental purposes, this square wave should have a very rapid rise and fall time, as this reduces the number of parameters to be considered in interpreting the results. This requirement, coupled with the high voltages and currents required to drive the luminaire, meant that a high-power, high-performance power supply was selected [31].

## 4. Position Estimation Algorithm

The position estimation algorithm used in this paper is explained with reference to Figure 6 and Figure 7. Figure 6 shows the light from the luminaire at position $\left(x_{\mathrm{RL}},y_{\mathrm{RL}},z_{\mathrm{RL}}\right)$. This light reaches the QADA with AOA $\left(\psi ,\alpha \right)$, where ψ is the azimuthal angle and α is the polar angle. It then passes through the square aperture and forms a square light spot on the plane of the QPD. The overlap of the light spot on the QPD is shown in more detail in Figure 7. The QADA coordinate frame has an origin at the center of this aperture and is rotationally aligned with it. The position of the center of the aperture in the room-based coordinate frame is $\left(x_{\mathrm{RQ}},y_{\mathrm{RQ}},z_{\mathrm{RQ}}\right)$. The function of the QADA is to estimate $\left(\psi ,\alpha \right)$, but as a first step, it estimates the point $\left(x_{\mathrm{QS}},y_{\mathrm{QS}},z_{\mathrm{QS}}\right)$, which is the centroid of the light spot.

By applying basic geometry, it can be shown that
(1)$$\frac{x_{\mathrm{QS}}}{L}=\left\{\begin{array}{l} \;\;\;\;\;\;\left(\frac{A_{1}+A_{4}}{A_{2}+A_{3}}-1\right),\;-1<\frac{x_{\mathrm{QS}}}{L}\leq 0\\ \left(1-\frac{A_{2}+A_{3}}{A_{1}+A_{4}}\right),\;\;0<\frac{x_{\mathrm{QS}}}{L}<1 \end{array}\right.,$$
where $A_{j}$ is the area of the overlap of the light spot on the jth quadrant, and L is the length of each quadrant of the photodiode.

Similarly,
(2)$$\frac{y_{\mathrm{QS}}}{L}=\left\{\begin{array}{l} \;\;\;\;\;\;\left(\frac{A_{1}+A_{2}}{A_{3}+A_{4}}-1\right),\;-1<\frac{y_{\mathrm{QS}}}{L}\leq 0\\ \left(1-\frac{A_{3}+A_{4}}{A_{1}+A_{2}}\right),\;\;\;\;\;0<\frac{y_{\mathrm{QS}}}{L}<1 \end{array}\right..$$

The important point to note in (1) and (2) is that $x_{\mathrm{QS}}/L$ and $y_{\mathrm{QS}}/L$ depend on the *ratio* of the areas, not on their *absolute* values.

By making reasonable assumptions about the light source and by using the fundamental properties of the photodiode, it can be shown [19] that the photocurrent $i_{j}(t)$ in the *j*th quadrant at time t is given by
(3)$$i_{j}(t)=\frac{RP_{T}A_{j}(t)\cos^{2}\left(\psi (t)\right)}{\pi d^{2}(t)},$$
where d is the distance between the transmitter and the receiver, *R* is the responsivity of the PD, and $P_{T}$ is the transmitted optical power. As all of the terms in (3) except for $A_{j}\left(t\right)$ are independent of j (3) can be simplified to
(4)$$i_{j}(t)=KA_{j}(t),$$
where
(5)$$K=\frac{RP_{T}\cos^{2}\left(\psi (t)\right)}{\pi d^{2}(t)},$$

Substituting (4) in (1) and (2), it can be seen that K cancels out. This is an important advantage of the comparative approach (that is, between quadrants), which can be taken when using a QPD.

In practice, the estimates ${\hat{x}}_{\mathrm{QS}}$ and ${\hat{y}}_{\mathrm{QS}}$ are made using the sampled, digitized signals output from the ADC denoted by $r_{i}\left[n\right]$, rather than continuous signals. The algorithm first calculates the average of $N_{E}$ samples for each quadrant, where the average for the jth quadrant is given by $\mu_{j}=\sum_{n=1}^{N_{E}}r_{i}\left[n\right]/N_{E}$. The algorithm then calculates $\mu_{Hj}$, which is the average of the samples for which $r_{j}\left[n\right]>\mu_{j}$, and also $\mu_{Lj}$, which is the average for samples where $r_{j}\left[n\right]\leq \mu_{j}$. The use of averaging is important, both to reduce the effect of random noise and because the PSOC does not sample all quadrants precisely synchronously. The difference between $\mu_{Hj}$ and $\mu_{Lj}$ is then used as an estimate of the power of the square-wave light signal reaching the jth quadrant, and from this $\hat{A_{j}}$, an estimate of the relative area of overlap of the light spot on the jth quadrant is made as follows:(6)$$\hat{A_{j}}=\mu_{Hj}-\mu_{Lj}.$$

In the experiments, the luminaires were modulated with square waves, but because all quadrants receive signals of the same form, this algorithm can be used for other waveshapes. Also, $z_{\mathrm{QS}}=-h$, where h is the height of the aperture above the photodiode plane. The estimates $\left({\hat{x}}_{\mathrm{QS}},{\hat{y}}_{\mathrm{QS}}\right)$ are made by substituting (6) in (1) and (2). Now, converting to polar coordinates gives $\hat{\alpha }$ the estimate of α
(7)$$\left(\frac{\hat{r}}{L},\hat{\alpha }\right)\Leftrightarrow \left(\frac{{\hat{x}}_{\mathrm{QS}}}{L},\frac{{\hat{y}}_{\mathrm{QS}}}{L}\right).$$

In turn, the estimate $\hat{\psi }$ is given by
(8)$$\hat{\psi }=\arctan\left(\frac{L}{h}\times \frac{\hat{r}}{L}\right).$$

An important point to note is that $\hat{\alpha }$ does *not* depend on the height of the aperture or the size of the photodiode; whereas $\hat{\psi }$ depends on the ratio L/h.

For the special case considered in some of the later analyses, where $z_{\mathrm{RL}}$ and $z_{\mathrm{RQ}}$ are known and the three axes of the room-based coordinate system are parallel to the three axes of the QADA-based frame, the position of the QADA can be calculated directly from $x_{\mathrm{QS}}$ and $y_{\mathrm{QS}}$. Applying basic geometry gives
(9)$$x_{\mathrm{RQ}}=x_{\mathrm{RL}}+\frac{L}{h}\left(z_{\mathrm{RL}}-z_{\mathrm{RQ}}\right)x_{\mathrm{QS}},$$
and
(10)$$y_{\mathrm{RQ}}=y_{\mathrm{RL}}+\frac{L}{h}\left(z_{\mathrm{RL}}-z_{\mathrm{RQ}}\right)y_{\mathrm{QS}}.$$

## 5. Experimental Results

An extensive set of measurements were made in the test chamber using the QADA receiver. The results were used to calculate the accuracy with which the QADA prototype can estimate AOA and also to demonstrate that it can reliably identify the transmitting luminaire and form the first stage of a HIP receiver. In these experiments, the transmitting luminaire was a square luminaire positioned with a centroid at $\left(x_{\mathrm{RL}},y_{\mathrm{RL}},z_{\mathrm{RL}}\right)=\left(1755,1260,2413\right)$ mm. The QADA was mounted on the test platform at a height $z_{\mathrm{RQ}}=1025$ of mm, and measurements were taken at 81 different positions on a 250 mm grid. At each position, 10 data captures of 5000 values per quadrant were made. The resulting information was then processed in MATLAB, and the AOA estimates were calculated using the algorithm described above. These estimates were compared with the corresponding ‘calculated’ values $\alpha_{C}$ and $\psi_{C}$, which were calculated using the coordinates of the centroid of the luminaire and the position of the platform, and assuming that the QADA was perfectly positioned and aligned with the floor grid. If there is any error in the positioning of the QADA or measurement of the centroid of the luminaire, the calculated values will not be exactly equal to the actual values.

### 5.1. Angle-of-Arrival Measurements

The formulae derived in the previous section for the polar and azimuthal angles were applied to the captured data. Figure 8a plots the estimated value of the polar angle $\hat{\alpha }$ versus the calculated value $\alpha_{C}$ for the 810 captured sequences. The red line shows the ideal result where $\hat{\alpha }=\alpha_{C}$. For most positions, there is very close agreement. The error $\hat{\alpha }-\alpha_{C}$ was large in only two positions: for $\left(x_{\mathrm{RQ}},y_{\mathrm{RQ}}\right)=\left(2000,1250\right)$ and for $\left(x_{\mathrm{RQ}},y_{\mathrm{RQ}}\right)=\left(1750,1250\right)$. At these positions $x_{\mathrm{QS}}/L\approx 0$ and/or $y_{\mathrm{QS}}/L\approx 0$, and a small error in the estimation of the centroid of the spot, resulted in the estimated position being in the incorrect quadrant of the photodiode. The position $\left(x_{\mathrm{RQ}},y_{\mathrm{RQ}}\right)=\left(1750,1250\right)$ is directly under the luminaire, and position $\left(x_{\mathrm{RQ}},y_{\mathrm{RQ}}\right)=\left(2000,1250\right)$ is almost in line with the centroid of the luminaire. While these errors in $\hat{\alpha }$ may appear large, we show later that these errors do not significantly reduce the ability of the QADA to correctly identify the transmitting luminaire. Figure 8b is a graph of the error $\hat{\alpha }-\alpha_{C}$ excluding those two receiver positions. It can be seen that for most positions $\left|\hat{\alpha }-\alpha_{C}\right|<8$ degrees.

The results for the azimuthal angle are shown in Figure 9, which compares the estimates $\hat{\psi }$ with the calculated values $\psi_{C}$. As noted in the previous section, $\hat{\psi }$ depends on L/h. To derive these estimates, we measured an ‘effective’ height by processing the results using a number of different values of h and selecting the one which gave the best fit. In Figure 9, an effective aperture height of h=2.2 mm was used. Figure 9a shows quite close agreement between $\hat{\psi }$ and $\psi_{C}$. The red line shows the ideal result $\hat{\alpha }=\alpha_{C}$. Figure 9b, which plots the difference $\hat{\psi }-\psi_{C}$ against $\psi_{C}$, reveals that the magnitude of the error increases rapidly for $\psi_{C}>45$ degrees. These are the positions where the QADA is furthest from the luminaire. As discussed in the next section, this may be caused by refraction through the silicone resin, which covers the photodiode.

### 5.2. Identification of Transmitting Luminaire

When the QADA is used as the first stage in a HIP receiver, the key factor is whether the angle estimates are accurate enough for the transmitting luminaire to be correctly identified. To demonstrate that this can be achieved, the position of the luminaire was predicted using the estimated angles, the position of the QADA, and the height of the luminaire above the plane of the QADA. The predicted positions (red dots) are shown in Figure 10a, which also shows the outline of the square luminaire (blue lines) and the position of the luminaire within the test chamber. The predicted position is within the luminaire outline for every capture and for every QADA position. In other words, the QADA correctly identifies the luminaire. Figure 10b expands the detail around the luminaire.

It is clear that while all of the predictions are within the area of the luminaire, they are not precisely on the centroid. There is clearly also some structure to the errors. There are three small clusters of dots relatively far from the luminaire centroid. For these, the error is a result of the larger error in $\hat{\psi }$ for larger $\psi_{C}$. The main cluster, although it is near the centroid, is clearly offset from it. As discussed in the next section, this is probably due to the limitations of the aperture.

## 6. Discussion of Experimental Results and Possible Sources of Error

While the experiments have demonstrated that a QADA can provide good AOA estimates and can consistently identify the transmitting luminaire, its performance is not perfect. It is useful to explore the causes of the errors, both to guide the construction of improved QADAs to be used in HIPs and to pave the way to improved QADAs being used in combination with other sensors to create novel positioning systems requiring only one or a small number of transmitting luminaires.

In the experiments, the three axes of the room-based coordinate system are parallel to the three axes of the QADA-based frame, and $z_{\mathrm{RL}}$ and $z_{\mathrm{RQ}}$ are known, so (9) and (10) can be used to predict the QADA positions. Figure 11 shows the predicted and actual positions of the QADA. It can be seen that for most positions, the predicted positions are very close to the actual positions. The exceptions are for the positions near the top left corner of the figure, which are the positions furthest from the luminaire and, therefore, where ψ is the largest. For most other positions, while the error is small, the estimated position is slightly to the right of the actual position. In other words, ${\hat{x}}_{\mathrm{RQ}}>x_{\mathrm{RQ}}$ and also ${\hat{y}}_{\mathrm{RQ}}<y_{\mathrm{RQ}}$, although the error in the y direction is, on average, much smaller.

Figure 12a,b are also useful in interpreting the results. Both, like Figure 10 in the previous section, show the predicted position of the luminaire centroid, but in these, some calculated values rather than estimated values of the angles are used. Figure 12a shows the predicted position using the calculated value $\psi_{C}$ and the estimated value $\hat{\alpha }$, while Figure 12b uses the calculated values $\alpha_{C}$ and the estimated values $\hat{\psi }$. It is clear that errors in both $\hat{\alpha }$ and $\hat{\psi }$ contribute to the errors in estimating the luminaire position but that the errors in $\hat{\psi }$ contribute more and, in particular, are the main cause of the three outlying clusters.

### 6.1. Errors Due to QADA Design and Construction

We now consider whether the design and construction of the QADA contributed to these errors. We have shown in Section 2 that the PSOC and the QPD without an aperture are very well matched. We now consider the possible source of errors when the aperture is fitted.

The photodiode that was used is a surface-mount type in a ceramic chip carrier with the QPD embedded in transparent silicone resin [29]. According to the data sheet, the overall thickness of the resin and the ceramic base is 1.26±0.15 mm, and the photosensitive surface is 0.46 mm below the surface of the resin. This means that the distance from the top of the aperture to the photosensitive surface is not precisely defined. This is why an ‘effective’ height was used to calculate the results in the previous section.

The refractive index of the resin is not specified, but a typical value for a polymer is 1.5. This means that for large angles of incidence, the resin will cause significant refraction, and the centroid of the light spot will be moved toward the center of the QPD. The change in $\hat{r}/L$ and hence on $\hat{\psi }$ depends on both the thickness of the resin and the distance from the top of the aperture to the top of the resin.

Ideally, the aperture is exactly the same size as the QPD and is precisely aligned with it. To determine whether this was true for the QADA used in the experiments, measurements were made with the QADA at a single location $\left(x_{\mathrm{RQ}},y_{\mathrm{RQ}},z_{\mathrm{RQ}}\right)=\left(1000,1000,1025\right)$, but it was rotated clockwise by 90 degrees between each set of 10 captures. The transmitter was the same as in the rest of the experiments. The average of the 10 captures for a given rotation for each quadrant is given in Table 1.

The top row gives the average for the quadrant directly opposite the luminaire for each of the four rotations. The relative position of the luminaire and QADA are the same as the example of the light spot shown in Figure 6. It can be seen from Figure 7 that for this configuration, the light spot completely covers the quadrant, and in the ideal case, the power of the measured signal is proportional to $L^{2}$. The table shows that the outputs from each of the four quadrants are very closely matched for this position: the difference between the largest and smallest value in the top row is a fraction of a percent. From Figure 7, we would expect the quadrant closest to the luminaire to have the smallest signal. This is consistent with the results shown in the bottom row of Table 1.

Detailed consideration of the results for all four orientations indicates that the aperture is slightly too large on the side shared by Q1 and Q4. This would result in an underestimate of ${\hat{x}}_{\mathrm{QS}}$, particularly when $x_{\mathrm{RL}}>x_{\mathrm{RQ}}$; this is for those positions to the left of the luminaire in Figure 11. Similarly, the results in Table 1 indicate that the aperture is slightly too large on the side shared by Q3 and Q4 and suggest that this is the cause of the errors in ${\hat{y}}_{\mathrm{QS}}$. This is consistent with the results in Figure 11.

Any reflections of the modulated light from the inner edges of the aperture onto the QPD can also cause errors. This was not observed in these experiments, but it clearly had an effect in earlier experiments with an aperture printed using white filament rather than black.

The limitations of our simple prototype QADA are all related to aspects of the QPD and the aperture and could potentially be removed in an integrated QADA constructed for commercial application. The ‘effective’ aperture height is unknown but is fixed and, if known, can be corrected in the signal processing algorithms used to estimate the AOA. Similarly, the refraction that causes the large errors in $\hat{\psi }$ is fixed and predictable once the QADA is manufactured, and this could be mitigated by appropriate signal processing.

The mismatch between the aperture and the QPD was the most significant source of error in our experiment, and this is something that would be difficult to compensate for in software, but the mismatch could be substantially reduced by using a better 3D printer to manufacture the aperture. It could probably be virtually eliminated if the QPD and the aperture were constructed as one item. The technique of rotating the QADA, which was used in deriving Table 1, also provides a simple and effective way of checking the match between quadrants in any future prototype QADAs.

### 6.2. Errors Due to Test Environment

There are also a number of aspects of the test environment that could result in errors in the angle estimation. An obvious one is that the transmitting luminaire is a distributed, not a point source. A number of simulations were performed in which the transmitter was modeled as an array of Lambertian transmitters. Surprisingly, this did not change the estimated AOA significantly. This is consistent with Figure 11, where the errors in the position estimate close to or beneath the luminaire are not significantly different from other more distant positions. The observed errors are consistent with the aperture misalignment noted above.

Another possible source of error is modulated light from the luminaire being reflected from the walls of the test chamber onto the QADA. Again, there is nothing in our experimental results that suggests that this is a significant effect.

Having described these limitations, it is important to stress that the experimental results show that even with these imperfections, the QADA was able to provide good AOA estimates and reliably identify the position of the luminaire.

## 7. Future Research Directions Using a QADA

The accurate low complexity compact QADA that we have developed, combined with the flexibility and power of the PSoC, opens up a multitude of research opportunities. These include making minor improvements to the current design, implementing other receiver functions within the PSoC, and crucially combining the QADA with other sensors to provide a robust, accurate, and scalable VLP system.

The QADA performance could be improved simply by optimizing the parameters of the components within the PSOC. In the experiments, the average of 5000 samples from each quadrant was used to estimate the power of the light received by that quadrant, but many fewer samples could be used, and this would reduce the time taken to provide an AOA estimate with possibly a small trade-off in accuracy. The settings for the TIAs could be changed to increase the bandwidth, the resolution of the ADC could be reduced, and the multiplexing could be changed to reduce the switching time. More sophisticated signal processing algorithms could be used in the estimation.

There are many other functions required in a fully operational QADA. These include demultiplexing of signals from different transmitters, automatic gain control (AGC) to adapt for different levels of ambient light and power of received signals, synchronization, demodulation and decoding of transmitted data, mitigation of side-effect modulation in luminaires [32], etc. Fortunately, it should be possible to implement most of these within the PSoC. In the experiments, the parameters of the components within the PSoC were kept constant, but many of the parameters can be changed in software, and so can be adjusted in real time. For example, the gains of the TIAs could be reduced to accommodate larger signals, and the values of the current sources adjusted to compensate for slowly varying ambient light. ASCII data can be decoded within the PSoC. The QADA receiver provides more information than an RSS receiver about the components of a multiplexed signal, and this may be used, possibly with iterative algorithms, to improve the separation of the signals from different transmitting luminaires. The simple AOA estimation algorithm that we have described can be used with more advanced modulation schemes, such as asymmetrically clipped orthogonal frequency division multiplexing (ACO-OFDM), and it may be possible to demodulate these within the PSoC [33].

Because of the unavailability of a simple, accurate AOA sensor for either RF or visible light IPS systems, there is relatively little directly relevant literature on sensor fusion. As a result, this new QADA design opens up the prospect of many new VLP systems that combine AOA estimation with inputs from other sensors. The positioning capabilities of these systems will depend on the type of sensors, how many luminaires are within the FOV of the receiver, and whether the receiver is moving. The relationship between the number of transmitting beacons with the FOV of the receiver, prior knowledge about the receiver orientation, and the ability of the receiver to calculate position in 2 or 3 dimensions, as well as to estimate its orientation, are discussed in [34].

When only a single beacon is within the FOV, even if the receiver can only move in a plane parallel to the beacon, there is still a four-fold ambiguity in the estimated position. The addition of a simple, readily available magnetometer compatible with and controlled by the PSoC could resolve this ambiguity and provide accurate positioning within a plane. This could be used in the important case of mobile robots moving around the floor of a room. The addition of a magnetometer and tilt detectors would allow positioning in more dimensions. For a moving receiver, measurements taken at different receiver positions could be used to improve the accuracy of estimates. For scenarios where more luminaires are within the FOV of the receiver, even more sophisticated signal processing can be used to further improve position estimation.

There is no reason why the QADA cannot be combined with information from other sensors, including accelerometers, cameras, RF receivers of all types (WiFi, Bluetooth, UWB, GNSS, 5G), ultrasonic receivers, cameras, and ambient light sensors. Many of the approaches already used in IPS could also be applied, including fingerprinting and odometry. The QADA could also be used for outdoor positioning, including vehicular applications [35] and even underwater positioning [36].

## 8. Conclusions

The detailed design of a QADA AOA sensor for VLP has been presented. The new design requires only three parts: a quadrant photodiode, a PSoC, and a 3D-printed aperture, so the new design can readily be reproduced by other researchers. Extensive experimental results demonstrate that the new design can provide accurate AOA estimates within a 3 m × 3 m test chamber and can reliably identify the direction of the transmitting luminaire. The inaccuracy of the 3D-printed aperture is the main limitation of the current prototype, and it is anticipated that the use of better 3D printing will substantially reduce the estimation errors. The new design opens up many avenues for future research, which are outlined in the paper. They include implementing receiver functions such as demultiplexing and AGC, but perhaps more importantly, the flexibility of the PSoC interface will allow other types of sensors to be readily added to the system and form an accessible experimental platform for sensor-fusion research.

## Figures and Tables

**Figure 1 sensors-24-06006-f001:**
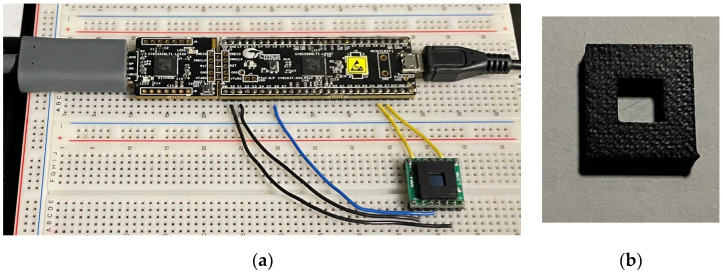
(**a**) QADA built on a prototyping board; (**b**) close-up of aperture removed from QPD.

**Figure 2 sensors-24-06006-f002:**
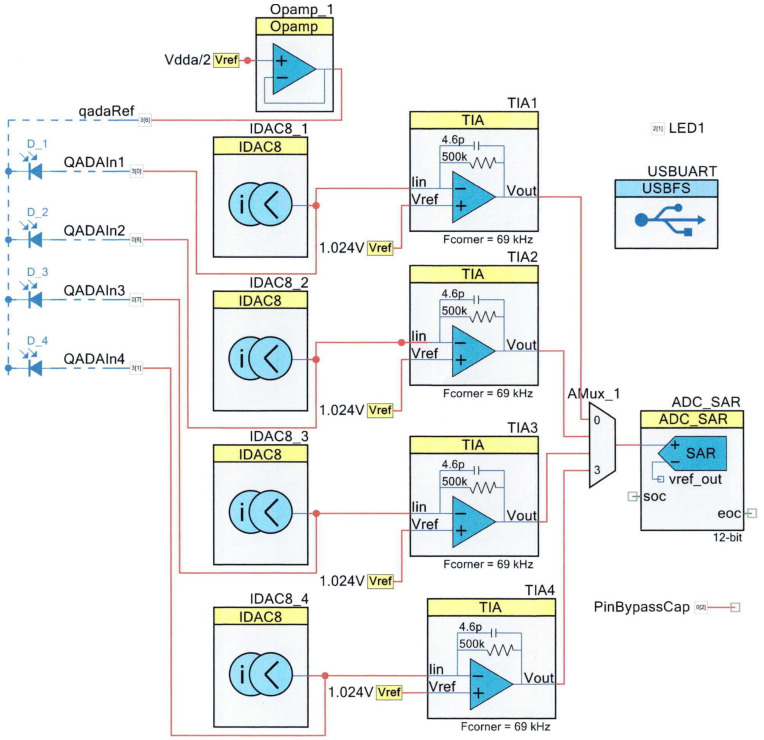
Circuit programmed in PSoC and external QPD.

**Figure 3 sensors-24-06006-f003:**
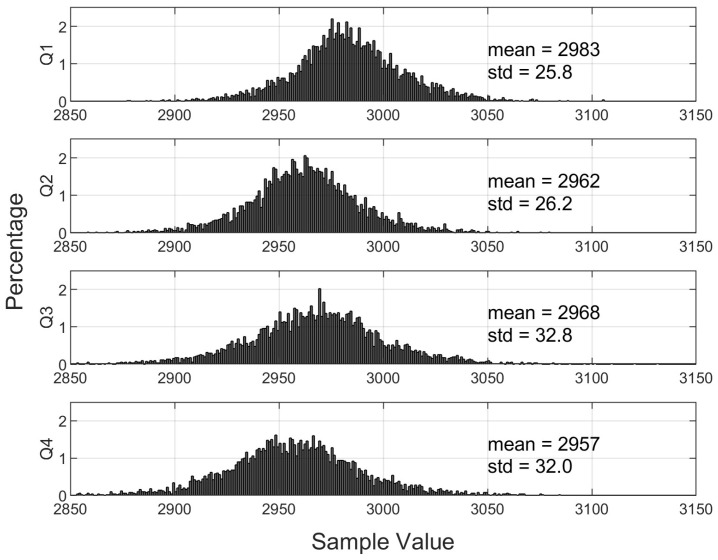
Histograms showing the distribution of samples for each quadrant for a QPD with no aperture and with no transmitting light.

**Figure 4 sensors-24-06006-f004:**
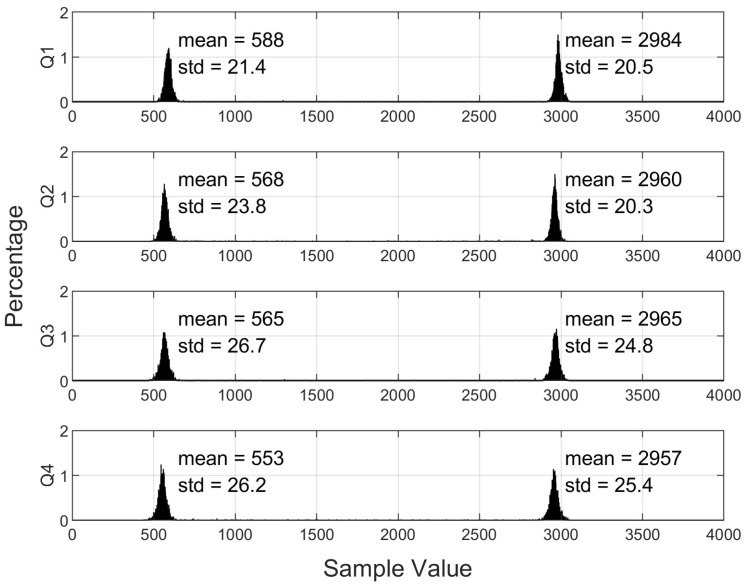
Histograms showing the distribution of samples for each quadrant for the case of no aperture and square-wave modulated light source.

**Figure 5 sensors-24-06006-f005:**
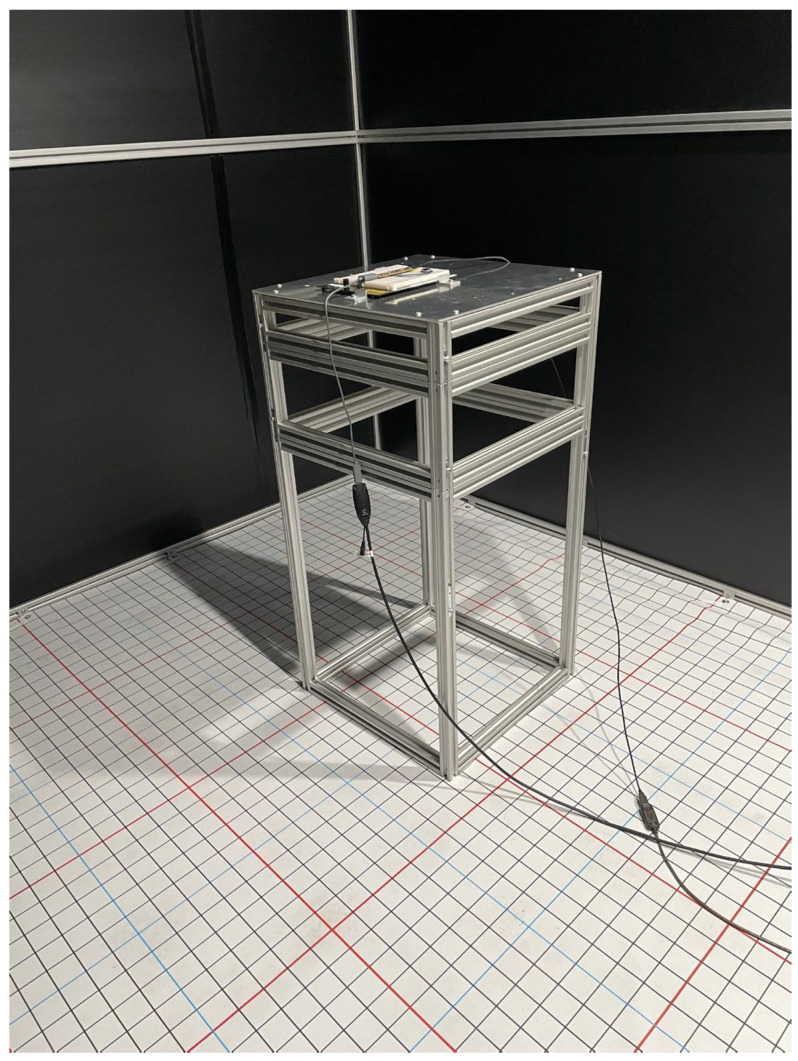
QADA prototype mounted on test platform in test chamber.

**Figure 6 sensors-24-06006-f006:**
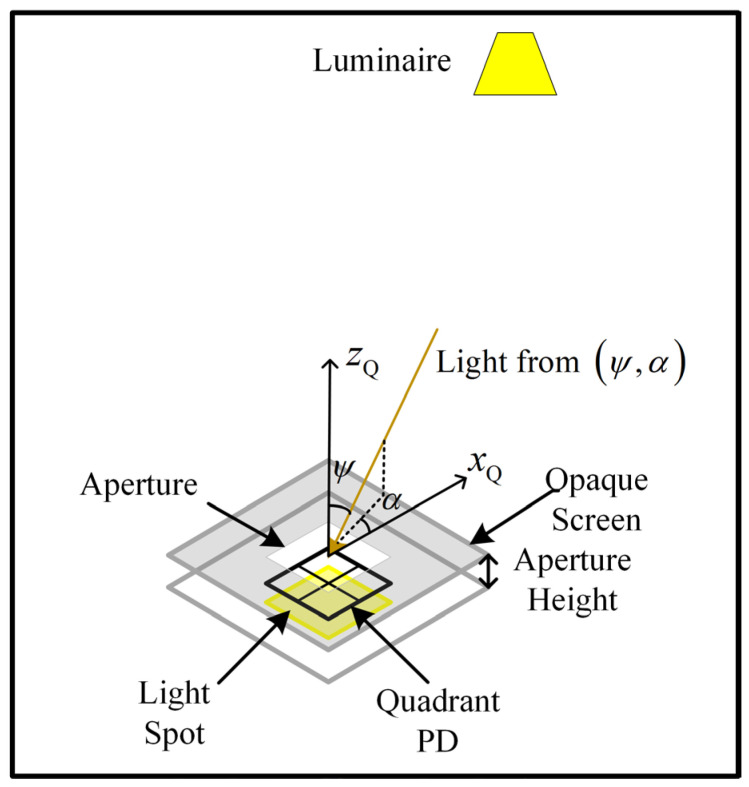
QADA receiver design.

**Figure 7 sensors-24-06006-f007:**
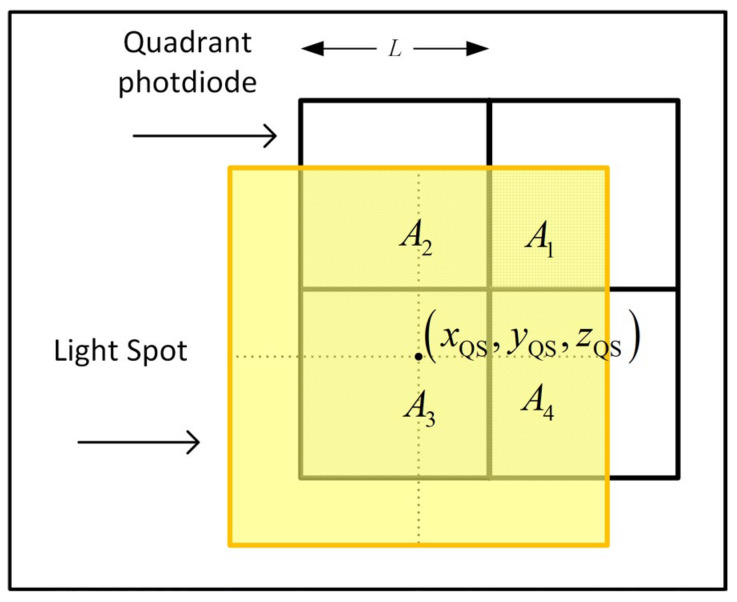
Detail of the light spot on the quadrant photodiode.

**Figure 8 sensors-24-06006-f008:**
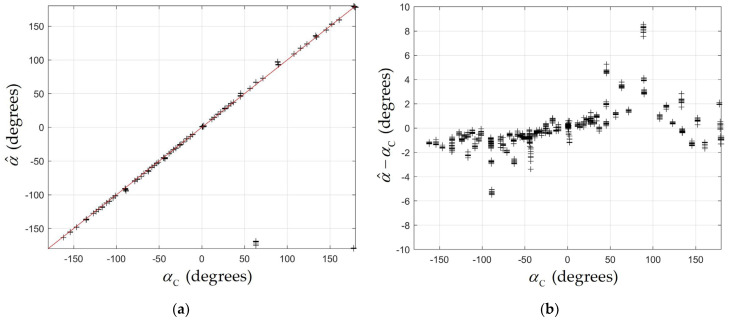
(**a**) Estimated angle $\hat{\alpha }$ versus calculated angle $\alpha_{C}$ (black crosses) and $\hat{\alpha }=\alpha_{C}$ (red line); (**b**) error in estimated angle $\hat{\alpha }-\alpha_{C}$ versus calculated angle $\alpha_{C}$ excluding outliers.

**Figure 9 sensors-24-06006-f009:**
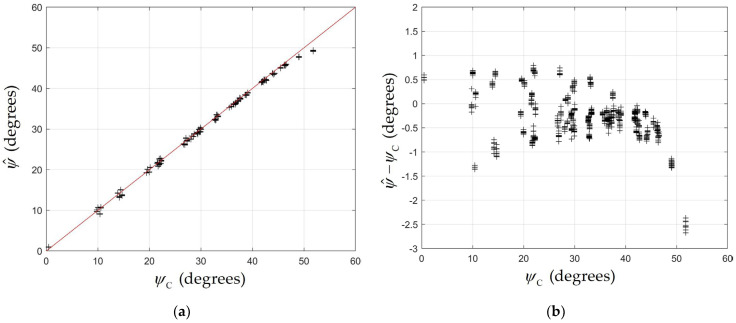
(**a**) Estimated angle $\hat{\psi }$ versus calculated angle $\psi_{C}$ (black crosses) and $\hat{\psi }=\psi_{C}$ (red line); (**b**) error in estimated angle $\hat{\psi }-\psi_{C}$ versus calculated angle.

**Figure 10 sensors-24-06006-f010:**
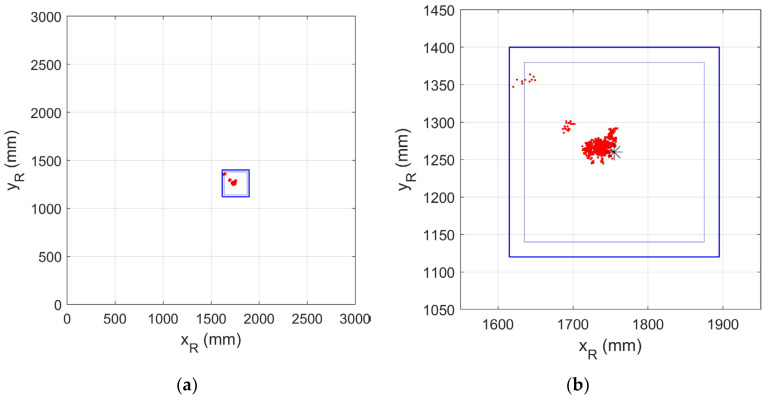
(**a**) Predicted positions within the room of the luminaire centroid (red dots), luminaire outline (blue lines); (**b**) predicted positions of luminaire centroid (red dots), luminaire outline (blue lines) and luminaire centroid (black asterisk) on an expanded scale. The inner blue line marks the area of the luminaire which transmits light. The outer blue lines include its metal frame.

**Figure 11 sensors-24-06006-f011:**
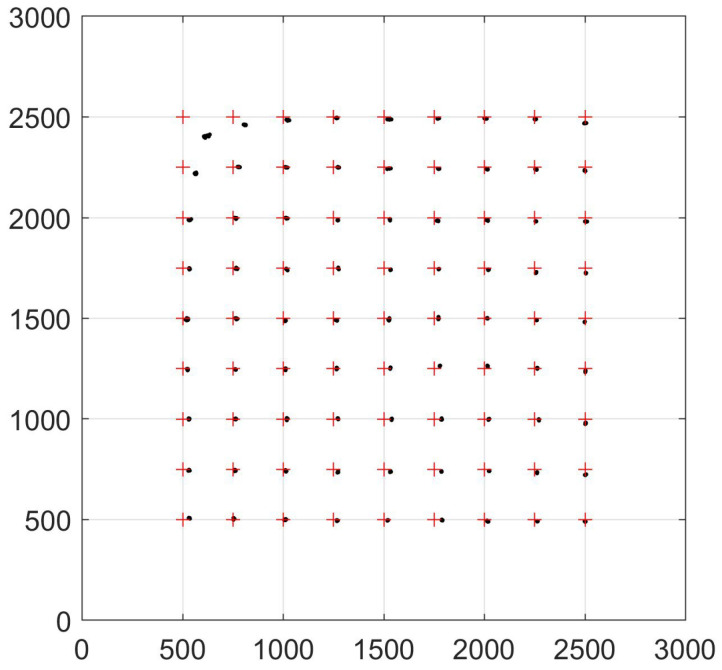
Predicted positions (black dots) and actual positions of QADA (red crosses). Predicted positions calculated using (9) and (10). The luminaire outline is shown in blue.

**Figure 12 sensors-24-06006-f012:**
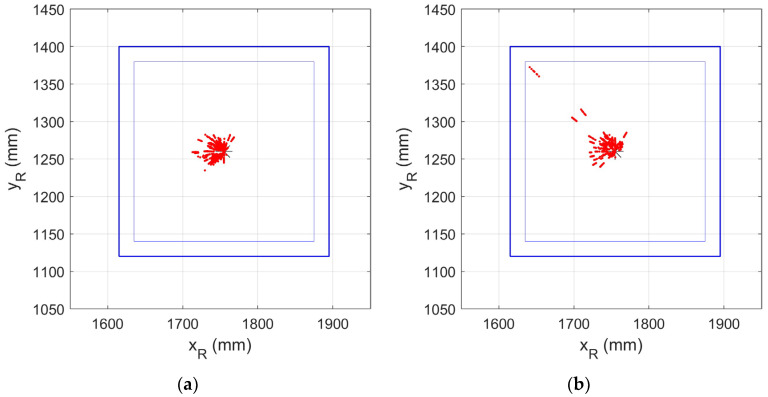
(**a**) Position of luminaire centroid predicted using $\psi_{C}$ and $\hat{\alpha }$; (**b**) position of luminaire centroid predicted $\hat{\psi }$ and $\alpha_{C}$.

**Table 1 sensors-24-06006-t001:** Average of signal amplitude on each of the four quadrants when QADA is rotated.

Quadrant Position	0° Rotation	90° Rotation	180° Rotation	270° Rotation	% Variation
Bottom Left	1380	1381	1381	1373	0.6%
Top Left	1151	1142	1162	1204	5.4%
Bottom Right	742	710	675	729	9.9%
Top Right	603	567	584	604	6.5%

## Data Availability

The raw data supporting the conclusions of this article will be made available by the authors upon request.

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
