# Peer review of "Accurate Low Complexity Quadrature Angular Diversity Aperture Receiver for Visible Light Positioning"

_sensors, 2024, doi:10.3390/s24186006_

Round 1
Reviewer 1 Report
Comments and Suggestions for Authors
This paper designs a new quadrant photodiode angular diversity aperture receiver for visible light positioning. Experimental results verify the superiority of the designed receiver. My comments are provided as follows:
1. Line 266, the authors stated that “Substituting (4) in (1) and (2)”. However, eq. (4) is the photocurrent, and eqs. (1) and (2) are positions. How to substituting (4) into (1) and (2)?
2. Lines 283-285, some symbols are not displayed correctly. Please correct them.
3. Lines 288, z_RL and z_RQ are assumed to be known. If z_RL and z_RQ are unknown, how to obtain the 3D positions of the receiver?
4. In experimental results section, the performance of the designed positioning receiver should be compared with the existing positioning receiver.
5. Some references on VLP are missing. For example,
B. Zhu, et al., “Three-dimensional VLC positioning based on angle difference of arrival with arbitrary tilting angle of receiver,” IEEE Journal on Selected Areas in Communications, 2018, 36(1): 8-22.
Reviewer 2 Report
Comments and Suggestions for Authors
The authors presented a detailed study on sensor technology, specifically focusing on improving the accuracy and performance of a system that uses light for detecting the Angle of Arrival (AOA) in visible light communication (VLC) systems. The research explores the design, implementation, and testing of a Quadrant Angular Diversity Aperture (QADA) that works with a Quadrant Photodiode (QPD) to enhance the precision of angle detection in VLC systems. However, the following comments and limitations should be considered:
· Experimental Design:
The experiments are well-structured and methodologically sound, providing clear evidence for the system's performance. However, there is limited information on the reproducibility of the experiments, which could be enhanced by including more details about the experimental setup and conditions. The sample size used in the experiments should be clarified. If the sample size is small, the statistical power might be insufficient to draw strong conclusions.
· Data Collection:
The data collection process appears to be rigorous, with multiple trials to ensure the reliability of the results. However, the paper lacks a discussion on potential sources of bias in data collection and how they were mitigated. It is important to ensure that data collection tools and methods are consistently applied across all trials. The paper would benefit from a more detailed explanation of these tools and methods.
· Data Analysis:
The data analysis section is comprehensive and uses appropriate statistical methods to interpret the experimental results. However, there could be more emphasis on the selection of these methods and why they are suitable for this particular study. The use of statistical significance testing is appropriate, but the paper should also consider the practical significance of the results. For example, how do the observed effects translate into real-world improvements in sensor technology? The paper could be strengthened by including a discussion of potential confounding variables and how they were controlled or accounted for in the analysis.
· Visualization and Interpretation:
The visualizations (graphs, tables) used to present the data are clear and effectively communicate the results. However, some visualizations could be more informative if additional context or comparison data were included. Interpretation of the data is logical and aligns with the research questions. However, the discussion could delve deeper into the implications of the findings and how they contribute to the broader field of sensor technology.
· Reporting of Results:
The results are reported accurately, but the paper could benefit from a more detailed reporting of the statistical parameters used, such as confidence intervals and effect sizes. It would also be useful to include a section on the limitations of the study, particularly in terms of experimental design and data analysis, and how these limitations might impact the generalizability of the results.
Round 2
Reviewer 1 Report
Comments and Suggestions for Authors
I have no more comments.
Reviewer 2 Report
Comments and Suggestions for Authors
The new version is better...